# Whole blood RNA-seq demonstrates an increased host immune response in individuals with cystic fibrosis who develop nontuberculous mycobacterial pulmonary disease

Miguel Dario Prieto[1,2], Jiah Jang[2], Alessandro N. Franciosi[1,2], Yossef Av-Gay[3], Horacio Bach[3], Scott J. Tebbutt[1,2,4], Bradley S. Quon[1,2]*

1 Department of Medicine, Faculty of Medicine, University of British Columbia, Vancouver, British Columbia, Canada, 2 Centre for Heart Lung Innovation, University of British Columbia and St Paul's Hospital, Vancouver, British Columbia, Canada, 3 Division of Infectious Diseases, Faculty of Medicine, University of British Columbia, Vancouver, British Columbia, Canada, 4 Prevention of Organ Failure (PROOF) Centre of Excellence, Vancouver, British Columbia, Canada

* bradley.quon@hli.ubc.ca

**Data Availability Statement:** The RNA sequencing data, raw reads, and normalized counts were deposited in the Gene Expression Omnibus

## Abstract

### Background

Individuals with cystic fibrosis have an elevated lifetime risk of colonization, infection, and disease caused by nontuberculous mycobacteria. A prior study involving non-cystic fibrosis individuals reported a gene expression signature associated with susceptibility to nontuberculous mycobacteria pulmonary disease (NTM-PD). In this study, we determined whether people living with cystic fibrosis who progress to NTM-PD have a gene expression pattern similar to the one seen in the non-cystic fibrosis population.

### Methods

We evaluated whole blood transcriptomics using bulk RNA-seq in a cohort of cystic fibrosis patients with samples collected closest in timing to the first isolation of nontuberculous mycobacteria. The study population included patients who did (n = 12) and did not (n = 30) develop NTM-PD following the first mycobacterial growth. Progression to NTM-PD was defined by a consensus of two expert clinicians based on reviewing clinical, microbiological, and radiological information. Differential gene expression was determined by DESeq2.

### Results

No differences in demographics or composition of white blood cell populations between groups were identified at baseline. Out of 213 genes associated with NTM-PD in the non-CF population, only two were significantly different in our cystic fibrosis NTM-PD cohort. Gene set enrichment analysis of the differential expression results showed that CF individuals

repository of NCBI (https://www.ncbi.nlm.nih.gov/geo/query/acc.cgi?acc=GSE205161, access token: kxsteiucbhebbkp). The dataset will be publicly available without restriction once we finish the publication process. The scripts for analysis of clinical, demographic, and genomic data have been deposited with the rest of the necessary data sets in dryad (https://datadryad.org/stash/share/FZ1DbrVy1KzveU5UQ_UbjqG-IrfFSouFSpspXhdnghg). While completing the submission process, the data is accessible by request of the reviewers. Once published, it will be made available without restrictions.

**Funding:** Franciossi AN is supported by a Research Trainee Award (RT-2020-0493) from the Michael Smith Foundation for Health Research (https://www.msfhr.org/). Quon BS is supported by a Michael Smith Foundation for Health Research Scholar Award, with no reference number. Av-Gay Y is supported by the Cystic Fibrosis Canada Foundation, with no reference number (https://www.cysticfibrosis.ca/). The funders had no role in study design, data collection, and analysis, decision to publish, or preparation of the manuscript.

**Competing interests:** The authors have declared that no competing interests exist.

who developed NTM-PD had higher expression levels of genes involved in the interferon (α and γ), tumor necrosis factor, and IL6-STAT3-JAK pathways.

## Conclusion

In contrast to the non-cystic fibrosis population, the gene expression signature of patients with cystic fibrosis who develop NTM-PD is characterized by increased innate immune responses.

## Introduction

Cystic fibrosis (CF) is a life-limiting genetic condition more prevalent in white populations (affecting approximately 1 in 3,000–4,000 live newborns) [1, 2]. People living with CF experience chronic respiratory symptoms and a steady decline in lung function leading to an increased risk of death or the need for a lung transplant [2, 3]. CF patients suffer from persistent and repetitive infections, including an increased lifetime risk of infection and disease caused by nontuberculous mycobacteria (NTM). According to data from the United States of America CF registry, around 12% of CF individuals in a given year have a positive culture for NTM, mostly of species belonging to the *Mycobacterium avium* (MAC) and *Mycobacterium abscessus* (MABs) complexes [3]. The clinical course of NTM infection in patients with CF is variable, ranging from an isolated positive culture to NTM pulmonary disease (NTM-PD) [4, 5]. The latter is diagnosed in a patient with two or more positive cultures of the same NTM, clinical deterioration, and characteristic radiological findings after excluding more common causes of the deteriorated status [4]. In CF, NTM-PD has been associated with an accelerated decline in lung function, and sputum positivity for NTM may affect eligibility for lung transplants [4–8].

Clinically, NTM infection in CF has been linked to specific age groups, clinical history of allergic bronchopulmonary aspergillosis, infection with *Aspergillus* spp., and chronic exposure to immunomodulatory drugs like macrolides and steroids [9–14]. However, fewer studies have evaluated risk factors for the progression from NTM infection to NTM-PD in the CF population. Caverly et al. showed that changes in the microbiome composition, including relative abundance of the *Rhotia* taxonomic unit, were associated with developing NTM-PD (n = 25) [15]. Furthermore, based on a prospective cohort of 96 CF patients with positive NTM culture, Martiniano et al. showed that those who progressed to NTM-PD had lower baseline Forced Expiratory Volume in 1 second (FEV1) and a faster lung function decline in the year before NTM infection [5]. While the role of the host immune response in the development of NTM-PD remains unclear in CF, various genetic polymorphisms have been reported in association with the diagnosis of NTM-PD in the non-CF population [16]. Additionally, Cowman et al. compared whole blood transcriptomics of patients with chronic obstructive pulmonary disease or non-CF bronchiectasis with and without a diagnosis of NTM-PD. Using microarray technology, they found reduced expression of genes involved in lymphocyte effector functions and interferon-γ production in those who developed NTM-PD. Also, they reported that those with NTM-PD and poor survival after diagnosis had a positive enrichment (higher expression) of innate immunity pathways compared to those with NTM-PD and better outcomes [17]. No prior studies have studied the relationship between host immune response to NTM and the development of NTM-PD in CF. In this exploratory study, we hypothesized that patients with CF who develop NTM-PD would have intrinsic downregulation of gene

expression in interferon-γ and immune-related pathways. Thus, we evaluated patterns of gene expression associated with NTM-PD in a cohort of CF patients using whole-blood transcriptomics.

## Methods

### Study population and clinical data

This study is a secondary data analysis using blood samples and data from the "CF Biomarker" cohort approved by the University of British Columbia-Providence Health Care ethics review board (H12-00835). The local ethics board also approved the secondary analysis (H20-00117). Patients in the parent cohort were recruited following informed consent at the St. Paul's Hospital Adult CF Clinic (Vancouver, Canada) between January 2012 and December 2019. In the current study, we included participants who consented to the future use of their samples and data, had at least one positive respiratory culture for NTM, and had a whole blood RNA sample available (PAXgene® stored at -70°C). Lung transplant recipients and subjects without a definite diagnosis of CF were excluded [18]. We preferentially selected blood samples taken during clinically stable periods and closest to the first positive growth of NTM. We did not limit blood sampling to within a specified time frame in relation to NTM infection or the development of NTM-PD as we expected an intrinsic downregulation of gene expression in immune-related pathways in patients with CF who develop NTM-PD.

### Clinical and demographic data analysis

Demographic characteristics, anthropometric measurements, pulmonary function tests, CFTR genotyping results, and microbiology laboratory reports were extracted from the clinical charts using a pre-defined case report form. Clinical and demographic data were extracted starting from 2 years before the recorded time of first NTM isolation for each patient (as far back as May 2001) and follow-up data was reviewed for the development of NTM-PD or not up until September 2022. Participants were censored at the time of NTM-PD diagnosis, lung transplant, or death. The diagnosis of NTM-PD was defined independently by two expert CF clinicians, based on current CF NTM guidelines, and disagreements were resolved by consensus [4]. Lung function measurements (forced expiratory volume in 1 second [$FEV_1$]) were standardized according to the 2012 Global Lung Function Initiative equation [19]. Clinical and demographic differences were compared according to the development or not of NTM-PD using univariable statistics in R studio (build 443) and R version 4.1.3 with the tidyverse package collection [20–22]. Statistical comparisons used parametric and non-parametric tests depending on the skewness of data and the type of variable analyzed, the significance level was defined at alpha $< 0.05$. Plots were produced using the ggplot and ggpubr packages [23, 24].

### RNA extraction and RNA sequencing experiments

The PAXgene Blood micro-RNA Kit was used for RNA extraction (in five batches) following the manufacturer's instructions (QIAGEN) and omitting the DNA depletion steps [25]. The quality of the extracted RNA was evaluated using the 260/280nm ratio in a NanoDrop™ spectrophotometer; the mean concentration was $117.4 \pm 73.2$ (SD) ng/uL. The RNA Integrity Number (RIN) was evaluated using a 2100 Bioanalyzer instrument from Agilent. A RIN $\geq 7$ was required for sequencing; two samples failed bioanalyzer quality control and were re-extracted. Poly-A RNA-seq libraries were prepared in a single batch with a starting concentration of 250 ng per sample. The library preparation was performed using the strand-specific Nextera NEB mRNA kit with adapters `AGATCGGAAGAGCACACGTCTGAACTCCAGTCAC` and

AGATCGGAAGAGCGTCGTGTAGGGAAAGAGTGT. The NovaSeq 6000 S4 PE100 (Illumina) platform with automatic base calling (RTA3) and an initial concentration of 200 pM per library was used for sequencing. Samples were multiplexed in a single flow cell after dual barcoding. The sequencing run generated 150 bp paired-end reads, the lowest mean Phred+33 score per file was 36/40. The median number of reads per sample was $65 \times 10^6$ (range: 36-151 $\times 10^6$). All transcriptomic data from this study can be found in NCBI Gene Expression Omnibus with accession GSE20516. The scripts to reproduce our results and additional metadata are deposited in the Dryad Digital Repository (doi:10.5061/dryad.np5hqbzx2).

## RNA-seq data analysis

Alignment and quantification of raw reads (FASTQ format) were performed in a High-Performance computational cluster (CentOS 7 Linux). Fast-QC was used to evaluate the quality of raw reads before and after alignment [26]. Untrimmed reads were aligned to the primary human assembly GRCh38.p13 v38 (May 21, 2021) from GENCODE using STAR v 2.7 [27, 28]. We produced gene-level quantification (bam format) in RSEM v1.3.3 with default parameters [29]. Quality control of alignment was performed using Picard tools, and quality reports were summarized with MultiQC [30, 31]. The raw count data was imported to R version 4.1.1 using tximport [32] and annotated with the Ensembl database version 104 of Bioconductor [33, 34]. Globin subunits and genes showing abnormally large expression counts ($>7^*10^7$) were filtered out before analysis; prior to filtering, 68–93% of reads per sample were overrepresented. An outlier sample was detected in exploratory principal component analysis (CFB2006) and removed from the dataset. Count data were corrected for extraction batches using the ComBat-seq algorithm [35]. Differential gene expression of the group with vs. without NTM-PD was performed using DESeq2 (v 1.3.2) with a cut-off FDR of 0.3. No minimum fold-change threshold was defined to increase the recovery of differentially expressed genes in this exploratory cohort [36]. Finally, gene set enrichment analysis of the molecular signatures database (human hallmark pathways) was performed with the fgsea package using the fold change ranked results from DESeq2 [37–40]. The cell-specific population compositions were estimated and explored using gene expression deconvolution in cibersortX against the complete blood cell counts (CBC) measured in patients at the time of sample procurement. The deconvolution process used the LM22 (22 blood immune cell types) signature matrix, bulk mode batch correction, and a hundred iterations [41]. Reporting of high-throughput sequencing methods follows the MINISEQE recommendations [42].

## Results

### Study cohort selection

Among 189 participants included in the CF biomarkers study, 53 had one or more positive airway cultures for NTM, and 42 of these fulfilled all eligibility criteria and were included in the study. Exclusion criteria included unclear CF diagnosis (n = 1), lung transplant recipient (n = 1), and no whole blood RNA sample available (n = 9). The diagnosis of NTM-PD was defined independently by two expert clinicians with a Cohen's kappa of 90%. Overall, 12 out of the 42 (29%) participants progressed to NTM-PD during clinical follow-up (up to September 2022). Among the 30 patients who did not develop NTM-PD, 22 spontaneously cleared the NTM from their respiratory cultures (12 after a single positive culture and 10 after multiple cultures), 3 had chronic colonization (NTM positive on all cultures) and 5 had intermittent positive cultures for NTM. The median follow-up time was 83.6 months (IQR 43–136), and the median interval between the first NTM growth and diagnosis of NTM-PD was 14.8 months (IQR 3–30). Patients who did not develop NTM-PD were followed for a minimum of

20.7 months after the first isolation of an NTM. Most patients were infected with *M. avium complex* (n = 22 in total; n = 7 progressed to NTM-PD) or *M. abscessus* complex (n = 12 in total; n = 5 progressed to NTM-PD). The remaining mycobacterial species included *M. gordonae* (n = 3), *M. fortuitum* (n = 2), *M. cosmeticum*, *M. peregrinum*, and *M. simiae* (n = 1 each), and none of these developed NTM-PD.

**Clinical and demographic characteristics of the cohort.** Included participants were predominantly male (n = 29, 69%), with a median age of 25 years, and 83% had at least one copy of F508del. Table 1 summarizes the demographic and clinical characteristics of the study cohort at baseline (first NTM growth). No differences at baseline were found in exposure to azithromycin (prior three months) or oral steroids (prior month) between patients who did vs. did not progress to NTM-PD. Furthermore, baseline clinical characteristics, including body mass index, $FEV_1$, and sputum microbiology, were not significantly different between the NTM-PD groups. No significant differences were found in co-infection rates at baseline for *Pseudomonas aeruginosa*, *Burkholderia* spp., *Stenotrophomonas maltophilia*, or *Aspergillus* spp. At baseline, there were also no differences in demographic or clinical factors (age, sex, body mass index, ppFEV1, genotype, or co-morbidities) between patients infected with MAC or MABs (S1 Table). The median time window between the first NTM growth and whole blood RNA collection was 20.8 months (range between -15.2 and 171 months as 5/42 samples were collected before first NTM growth). Six patients were sampled after therapy for NTM-PD (median of 49 months before sampling).

There were no significant differences at baseline in the absolute number of leukocytes, neutrophils, monocytes, or lymphocytes between NTM-PD groups (S1 Fig). A principal

**Table 1. Demographic and clinical characteristics of included patients at baseline, stratified by NTM-PD status.**

| | Total (n = 42) | No NTM-PD (n = 30) | NTM-PD (n = 12) |
|---|---|---|---|
| **Age—median (IQR)** | 28 (22–37) | 30 (23–37) | 24 (21.8–34.8) |
| **Females—n (%)** | 13 (31.0) | 9 (30.0) | 4 (33.3) |
| **Genotype—n (%)** | | | |
| F508del/F508del | 17 (40.5) | 11 (36.7) | 5 (41.7) |
| F508del/other | 18 (42.9) | 13 (43.3) | 6 (50.0) |
| Others | 7 (16.7) | 6 (20.0) | 1 (8.3) |
| **Pancreatic insufficiency—n (%)** | 33 (78.6) | 23 (76.7) | 10 (83.3) |
| **CF diabetes—n (%)** | 10 (23.8) | 7 (23.3) | 3 (25.0) |
| **Body mass index[a]—median (IQR)** | 22 (20.4–23.4) | 22 (20.7–23.6) | 22 (20.0–22.6) |
| **$FEV_1$ [% predicted][a]—median (IQR)** | 80 (63.7–89.3) | 80 (63.9–89.2) | 80 (61.6–89.5) |
| **Oral steroids exposure—n (%)[a]** | 3 (7.1) | 2 (6.7) | 1 (8.3) |
| **Macrolide exposure—n (%)[a]** | 14 (33.3) | 11 (36.7) | 3 (25.0) |
| **Follow-up time in months[b]—median(IQR)** | 83.6 (43–136) | 102 (75–139) | 14.8 (3–30) |
| **Sputum positivity—n (%)[c]** | | | |
| *Pseudomonas aeruginosa* | 15 (39) | 12 (41) | 3 (33) |
| *Aspergillus* spp. | 12 (32) | 8 (27) | 4 (44) |
| *Stenotrophomonas maltophilia* | 7 (18) | 4 (14) | 3 (33) |
| *Burkholderia* spp. | 3 (8) | 2 (7) | 1 (11) |

[a] n = 41, missing data for 1 patient.

[b] Follow-up time after first positive culture and until the patient developed NTM-PD, died, had a lung transplant, was lost to follow-up, or the study was closed (September 2022).

[c] n = 38, missing data for 4 patients.

IQR: interquartile range.

component analysis of all white blood cell parameters explained more than 50% of the variance in the first two components but did not separate NTM groups.

## Quality assessment of RNA-seq data

Quality control of raw reads did not show bias in nucleotide distribution, sequencing depth, or base calling quality. At least 94.6% of the reads per sample aligned to the reference genome, and at least 49.1% of them aligned to protein-coding regions, see S2 Fig. During exploratory principal component analysis of normalized and batch corrected gene counts, we found no particular grouping by extraction batch, biological sex, or mycobacterial species (after removing the CFB2006 outlier). A principal component analysis of normalized counts did not show a graphical difference according to the timing of sample procurement for RNAseq after dividing the cohort into three groups: before the first positive NTM culture, after the first isolation without progression to NTM-PD, and after the first isolation with progression to NTM-PD (S3 Fig). A PermANOVA test of this distance matrix mas not significant (p = 0.29) [43]. CybersortX deconvolution of sequencing counts showed a good prediction of the initial white blood cell counts. We observed Pearson correlations greater than 0.8 for lymphocyte and neutrophil counts and a moderate correlation (r = 0.4) for monocyte counts (S4 Fig).

## Evaluation of whole blood gene expression of candidate genes of NTM-PD in CF

We explored the candidate genes identified by Cowman et al. in our cohort. Among the 213 gene transcripts associated with NTM-PD in their cohort, only two genes (interleukin-2 receptor subunit Beta—IL-2RB and granzyme K—GZMK) were differentially expressed in our study population (FDR < 0.3) [17]. We also explored the mean expression of the differentially expressed genes with the highest fold change in the Cowman cohort. We found no apparent trends or statistical differences using univariable analysis. S5 Fig shows the mean expression of 13 of the top 15 features reported by Cowman et al. in our NTM-PD groups. We also evaluated the genes associated with poorer survival in those with NTM-PD among Cowman's cohort, as a potentially similar clinical course to the development of NTM-PD in CF. Still, none of the 215 genes associated with decreased survival was found as differentially expressed in our cohort according to the adjusted p-value. Furthermore, only 6 out of these 215 genes had an unadjusted p-value below 0.05.

## Whole blood gene expression of individuals who developed NTM-PD in our CF cohort

Differential gene expression with DESeq2 identified 111 differentially expressed genes (DEG) at an FDR cut-off of < 0.3. The top 30 DEG are summarized in Table 2 in descending order of adjusted p-value. The Bland–Altman, and Volcano plots in Fig 1 summarize the differential gene expression analysis results. We used gene set enrichment analysis of the molecular signatures database to explore the biological implications of these gene expression results. The results showed significant positive enrichment (higher expression in those who developed NTM-PD) in pathways involved in the interferon-α response (Adjusted p = 2.5 e$^{-09}$), interferon-γ response (Adjusted p = 2.5 e$^{-09}$), and IL6-JAK-STAT3 signaling (Adjusted p = 1.88 e$^{-05}$). Table 3 includes all enriched pathways with an adjusted p-value below 0.001.

**Table 2. Top 30 differentially expressed genes in those who did vs. did not develop NTM-PD.**

| SYMBOL | Adjusted p value | stat | DESCRIPTION |
|---|---|---|---|
| TBC1D3H | <0.001 | -10.849 | TBC1 domain family member 3H |
| TCL1A | 0.054 | 4.426 | TCL1 family AKT coactivator A |
| CD177 | 0.054 | 4.477 | CD177 molecule |
| NA | 0.054 | 4.504 | novel transcript, antisense to TCL1A |
| RN7SL731P | 0.069 | 4.322 | RNA, 7SL, cytoplasmic 731, pseudogene |
| ARHGEF25 | 0.070 | -4.279 | Rho guanine nucleotide exchange factor 25 |
| VPREB3 | 0.084 | 4.187 | V-set pre-B cell surrogate light chain 3 |
| FADS3 | 0.084 | 4.174 | fatty acid desaturase 3 |
| MKRN3 | 0.089 | -4.133 | makorin ring finger protein 3 |
| SCN3A | 0.095 | 4.053 | sodium voltage-gated channel alpha subunit 3 |
| CLEC17A | 0.095 | 4.067 | C-type lectin domain containing 17A |
| NA | 0.095 | 4.063 | novel transcript, antisense to HS3ST1 |
| PAK6 | 0.095 | -4.023 | p21 (RAC1) activated kinase 6 |
| NA | 0.095 | 4.016 | POM121 membrane glycoprotein-like 1 pseudogene |
| CACNA1C-AS1 | 0.109 | -3.967 | CACNA1C antisense RNA 1 |
| CTSW | 0.117 | -3.934 | cathepsin W |
| NA | 0.117 | -3.919 | novel transcript |
| NIBAN3 | 0.121 | 3.897 | niban apoptosis regulator 3 |
| DDX11L2 | 0.136 | -3.856 | DEAD/H-box helicase 11 like 2 (pseudogene) |
| RPL13P12 | 0.150 | 3.820 | ribosomal protein L13 pseudogene 12 |
| RBPMS2 | 0.158 | -3.784 | RNA binding protein, mRNA processing factor 2 |
| NA | 0.158 | -3.783 | novel transcript |
| RIMBP2 | 0.163 | 3.753 | RIMS binding protein 2 |
| NA | 0.163 | -3.753 | zinc finger protein 726 pseudogene 1 |
| GZMK | 0.176 | -3.660 | granzyme K |
| KLRC1 | 0.176 | -3.653 | killer cell lectin like receptor C1 |
| CLDN12 | 0.176 | -3.698 | claudin 12 |
| CMBL | 0.176 | 3.687 | carboxymethylenebutenolidase homolog |
| KRT72 | 0.176 | -3.720 | keratin 72 |
| SERF1A | 0.176 | 3.709 | small EDRK-rich factor 1A |

## Discussion

The main finding from this exploratory study is that CF individuals who developed NTM-PD after initial growth of NTM had increased expression of genes involved in pathways related to interferon responses, tumor necrosis factor-α production, and IL6-JAK-STAT3 signaling [17]. This contrasts with our expectation as we hypothesized that defective cellular immunity, as reflected by decreased gene expression in host immune response pathways, would confer an increased risk of persistent NTM infection and subsequent development of pulmonary disease.

While our results may seem contradictory to the findings reported by Cowman et al., which found defective T lymphocyte and interferon-γ immune responses in patients with NTM-PD, individuals with NTM-PD in their study were compared to control subjects with respiratory disease (COPD or bronchiectasis) but without NTM. As such, the decreased immune responses could have conferred susceptibility to NTM-PD in their study involving subjects without other known host immune-related risk factors for NTM. The airways of individuals with CF already create a hospitable environment for NTM growth and infection; therefore, reduced protective immunity in this setting could play a less significant role.

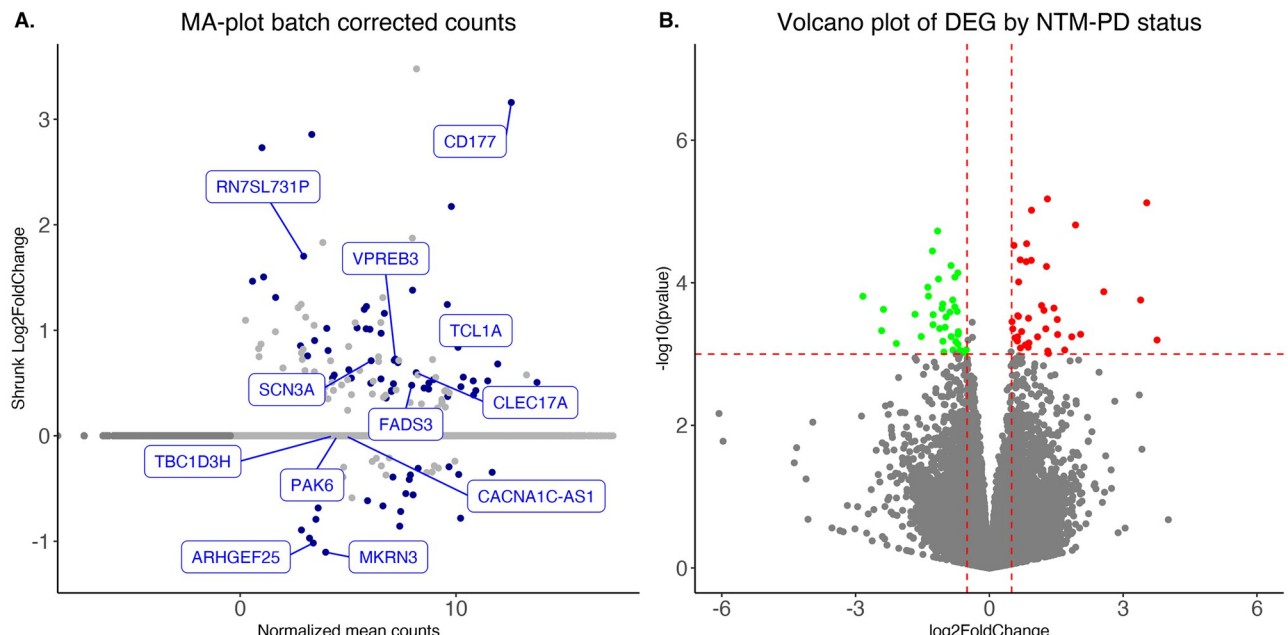

**Fig 1. Volcano plot and MA plot of DESeq2 differential expression analysis.** (A) Bland-Altman plot of normalized mean counts and shrunk (apeglm) Log2 fold-changes. The differentially expressed genes are in red, and the top 15 features are labeled. (B) Volcano plot of differential gene expression analysis with cut-offs of p< 0.001 and Log2 fold-change larger than absolute 0.5. Highlighted in red are genes with higher transcription levels in patients who progressed to NTM-PD, in green are those with lower expression in the same group.

In our study, we compared CF individuals with NTM infection who did vs. did not progress to NTM-PD, and a pro-inflammatory innate immune response characterized the latter. Indeed, much of the lung damage that occurs in CF results from the deleterious effects of an overexuberant inflammatory response to infection, as the host is incapable of controlling microbial growth due to intrinsic host defense defects related to CFTR dysfunction [44, 45]. Our results demonstrating increased immune activity in patients with NTM-PD are consistent with a recent study that compared the bronchoalveolar lavage fluid of CF patients with vs. without NTM infection and found that those with NTM infection had increased immune cells and inflammation [46]. Moreover, regions of the lung with increased inflammation correlated with greater lung tissue damage based on CT scans [46].

**Table 3. Gene set enrichment analysis of whole blood gene expression in those who did vs. did not develop NTM-PD.**

| Pathway | p-value | FDR | NES | Size |
|---|---|---|---|---|
| Interferon α response | <0.001 | <0.001 | 2.98 | 96 |
| Interferon γ response | <0.001 | <0.001 | 2.65 | 199 |
| IL-6-JAK-STAT3 signaling | <0.001 | <0.001 | 2.05 | 81 |
| Heme metabolism | <0.001 | <0.001 | 2.04 | 188 |
| Tumor necrosis factor-α signaling via NFK-β | <0.001 | <0.001 | 1.74 | 198 |
| Inflammatory response | <0.001 | <0.001 | 1.66 | 194 |
| Protein secretion | <0.001 | <0.001 | 1.53 | 86 |
| Oxidative phosphorylation | <0.001 | <0.001 | 1.51 | 188 |
| E2F targets | <0.001 | <0.001 | -1.60 | 193 |

Enrichment of molecular signatures database 3.0, accessed in May 2022. **NES**: normalized enrichment score, positive scores represent enrichment in patients that developed NTM-PD. **FDR**: false discovery rate.

Interestingly, several of the top differentially expressed genes found in the Cowman cohort were also downregulated in our cohort. Among them, granzyme K and interleukin 2 receptor β were differentially expressed in our cohort, see S5 Fig. Both genes are primarily expressed by lymphocytes but their relationship with susceptibility to NTM-PD remains unclear. Granzyme K is an effector protein involved in the cytotoxic activity of CD8+ lymphocytes, while interleukin 2 participates in the expansion of T cells. Despite the decreased expression of these genes, the pro-inflammatory response involving innate immune pathways related to interferons and tumor necrosis factor predominated in our enrichment results. The functional significance and the cells responsible for this increased production need to be further elucidated.

Our study has several limitations that should be acknowledged. The parent study was not initially designed for the discovery of new biomarkers or to evaluate the pathological process leading to NTM-PD in patients with CF. Instead, we performed a secondary exploratory analysis with existing samples to confirm if a gene expression pattern associated with NTM-PD was also present in CF patients. As such, we cannot ascertain the temporal relationship between infection with NTM and the gene expression profiles because the time window between NTM positivity and whole blood sampling was variable (median of 20 months). Based on preliminary data, we did not control for this factor as we expected to identify permanent intrinsic deficits in host defense associated with developing NTM-PD. Future studies will benefit from regular blood sampling in the period between the first NTM growth and the development of NTM-PD to determine if the innate immune response ramps up over time. Furthermore, additional factors that may affect the progression to NTM-PD, like the virulence factors of infecting NTM and the lung microbiome composition, should be explored. The gene expression results are unlikely to be biased by the initial proportion of cell populations because baseline CBC data showed no significant differences, and the deconvolution of gene expression data corroborated the distribution of cell populations in the CBC. The interval between the diagnosis of NTM-PD and sampling may also affect the results. Interestingly, a sensitivity analysis excluding 6 samples from patients sampled after starting treatment for NTM-PD displayed a similar pattern of enrichment to our main results (S2 Table). Finally, our cohort represents mostly patients naïve to CFTR modulator therapy as access to lumacaftor/ivacaftor and tezacaftor/ivacaftor was very limited during the study period in Canada due to lack of public and private reimbursement in most cases and elexacaftor/tezacaftor/ivacaftor was not reimbursed until September 2022. Only four patients, including one who progressed to NTM-PD, were receiving modulator therapy when the blood sample was collected (1 on ivacaftor, 2 on lumacaftor/ivacaftor, and 1 on tezacaftor/ivacaftor). As CFTR modulator therapy may change the lung microenvironment and affect the host susceptibility to mycobacterial infection and subsequent development of disease, our results may have limited generalizability to a CFTR modulator treated population but can serve as a baseline comparison for future studies in cohorts receiving these highly effective therapies [47].

In conclusion, our cohort shows that gene expression in patients who develop NTM-PD displays an exaggerated expression of genes that participate in innate immune responses. Overall, this study provides novel insights into the host response to NTM infection in the presence of CFTR dysfunction. The results of this exploratory cohort are valuable for future biomarker discovery and can serve as a validation dataset for other research groups.

## Supporting information

**S1 Table. Demographic and clinical characteristics of patients infected with MAC or MABs at the time of first positive NTM culture.**
(DOCX)

**S2 Table. Gene set enrichment analysis of whole blood gene expression in those who did vs. did not develop NTM-PD, excluding 6 samples taken after diagnosis of NTM-PD.** (DOCX)

**S1 Fig. Absolute whole blood cell population counts in samples used for RNAseq.** Boxplots with the distribution of absolute values for leukocytes, neutrophils, lymphocytes, and monocytes in complete blood counts at the time of blood sample procurement. No significant differences using the unpaired Wilcoxon rank-sum test. (TIF)

**S2 Fig. Assignment of reads to genomic regions produced by Picard tools.** Coding: protein-coding region. UTR: untranscribed region. Intronic: intronic region. Intergenic: intergenic region. Ribosomal: mapping to ribosomal RNA or proteins. PF not aligned: bases that passed the quality filter and were not aligned. (TIF)

**S3 Fig. Principal component analysis of normalized and batch corrected count data.** Separated by the timing of blood sample procurement for RNAseq into three categories: before first positive NTM culture, after first positive NTM culture without progression to NTM-PD, and after first positive NTM culture with progression to NTM-PD. No apparent separation is seen graphically. A PermANOVA test shows no significant differences in the distance matrices by group (p = 0.29). (TIFF)

**S4 Fig. CibersortX deconvolution of cell percentages compared to ground truth reference values in CBC.** (TIF)

**S5 Fig. Mean expression values per NTM-PD group in top differentially expressed genes in Cowman analysis.** The p-values represent the results of independent Wilcoxon tests. The feature **TRGC1** (T Cell Receptor Gamma Constant 1) was a duplicated probe in the initial analysis, while the uncharacterized locus **FLJ45825** was not detected in our dataset. **PMS2P1**: PMS1 homolog 2, Mismatch Repair System Component Pseudogene 1. **CRTAM**: Cytotoxic and regulatory T cell Molecule. **IFNG**: Interferon Gamma. **GZMK**: Granzyme K. **PZP**: PZP alpha-2-macroglobulin like. **XCL2**: X-C motif chemokine ligand 2. **AK5**: Adenylate kinase 5. **FCRL3**: Fc receptor Like 3. **PPIH**: Peptidylprolyl isomerase H. **A2M**: Alpha-2-macroglobulin. **TIGIT**: T cell immunoreceptor with Ig and ITIM domains. **MUC12**: Mucin 12, cell surface associated. (TIF)

## Acknowledgments

The authors would like to acknowledge the feedback and support of Kang Dong in the review of the bioinformatics pipeline employed for analysis. This research was supported in part through computational resources and services provided by Advanced Research Computing at the University of British Columbia.

## Author Contributions

**Conceptualization:** Miguel Dario Prieto, Yossef Av-Gay, Scott J. Tebbutt.

**Data curation:** Miguel Dario Prieto, Jiah Jang.

**Formal analysis:** Miguel Dario Prieto.

**Funding acquisition:** Bradley S. Quon.

**Investigation:** Miguel Dario Prieto, Jiah Jang, Alessandro N. Franciosi.

**Methodology:** Miguel Dario Prieto.

**Supervision:** Yossef Av-Gay, Horacio Bach, Scott J. Tebbutt, Bradley S. Quon.

**Writing – original draft:** Miguel Dario Prieto.

**Writing – review & editing:** Jiah Jang, Alessandro N. Franciosi, Yossef Av-Gay, Horacio Bach, Scott J. Tebbutt, Bradley S. Quon.

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
