## [Decision Letter · Decision Letter 0]

4 Oct 2022

PONE-D-22-24141Whole blood RNA-Seq demonstrates an increased host immune response in individuals with cystic fibrosis who develop nontuberculous mycobacterial pulmonary diseasePLOS ONE

Dear Dr. Prieto,

Thank you for submitting your manuscript to PLOS ONE. After careful consideration, we feel that it has merit but does not fully meet PLOS ONE’s publication criteria as it currently stands. Therefore, we invite you to submit a revised version of the manuscript that addresses the points raised during the review process.

We look forward to receiving your revised manuscript.

Kind regards,

Abdelwahab Omri, Pharm B, Ph.D, Laurentian University 

Academic Editor

PLOS ONE

Journal Requirements:

Reviewers' comments:

Reviewer's Responses to Questions

**Comments to the Author**

1. Is the manuscript technically sound, and do the data support the conclusions?

Reviewer #1: Yes

2. Has the statistical analysis been performed appropriately and rigorously? 

Reviewer #1: Yes

3. Have the authors made all data underlying the findings in their manuscript fully available?

Reviewer #1: Yes

4. Is the manuscript presented in an intelligible fashion and written in standard English?

Reviewer #1: Yes

5. Review Comments to the Author

Reviewer #1: This is a retrospective study evaluating whole blood gene expression using bulk RNA-seq in a cohort of cystic fibrosis patients with samples collected closest in timing to the first isolation of nontuberculous mycobacteria. The study population included patients who did (N=12) and did not (n-30) develop NTM-PD following first nontuberculous mycobacteria growth.

1. It is not stated in the inclusion/exclusion criteria that people on CFTR modulator therapy were excluded from analysis. However, this is mentioned in the discussion. Please add this to the exclusion criteria. Interestingly, the vast majority of patients were eligible for E/T/I therapy based on having one or more copies of F508del. I presume the timing of data abstraction preceded approval for E/T/I in Canada. Clarification of these details would be helpful, given most people meeting eligibility in the US (and some other locations) are likely on highly effective modulator therapy, skewing the presented results to a population that is not representative of most of North America. Please include some discussion of this issue in the manuscript.

2. Please better define the patients who did not develop NTM-PD. What percentage were intermittently colonized (NTM not recovered from every AFB culture, but rather intermittently) vs chronically colonized (NTM recovery with every AFB culture) vs spontaneously cleared infection (transient or chronic colonization, but clear after one year of negative cultures)?

3. Please define the data abstraction timeframe?

4. What was the total length of time that patients were followed to determine if they met criteria for NTM-PD? Some patients will be intermittently positive for years before conversion to NTM disease. Did some patients that were initially without NTM-PD found to have PD after data analysis was completed?

5. Per the manuscript, blood sampling was not limited to a specified timeframe in relation to NTM infection, but closest to the first positive growth. Presumptively these blood samples were collected exclusively after the first NTM recovery, but that is not explicitly stated. Please clarify. Additionally, 6/12 (50%) patients were sampled after initiation of NTM-specific therapy. Additional details surrounding the timing would be helpful.

6. In your analysis were there any differences between M. abscessus or M. avium patients? The n was very small, so statistical differences may not be observed. However, were trends observed? Please better define these data.

7. Table 1. Please define “follow up time”? Is this the total time a patient was followed post first + NTM culture?

Reference is missing in lines 270-273.

8. Have you compared bulk RNA-seq prior to first positive NTM isolate compared to the two groups?

Thank you for allowing be to participate in the review process.

6. PLOS authors have the option to publish the peer review history of their article (what does this mean?). If published, this will include your full peer review and any attached files.

Reviewer #1: No

---

## [Author Response · Author response to Decision Letter 0]

12 Nov 2022

Re: Manuscript PONE-D-22-24141

Dear Dr. Omri, 

We want to thank you and your group of reviewers for reviewing our manuscript “Whole blood RNA-Seq demonstrates an increased host immune response in individuals with cystic fibrosis who develop nontuberculous mycobacterial pulmonary disease” and providing meaningful feedback to improve it. We sought to address all the recommendations and provide clarifications when necessary. First, we addressed the journal requirements and then proceeded to respond to the comments posed by reviewer #1; the relevant changes associated to the reviewer comment are specified in the revised version of the manuscript without track changes.

Journal Requirements:

Response: We have reviewed the manuscript document and made the necessary corrections to the format and content to closely follow the requirements of PLOS ONE. The filenames of all documents, including figures and supplementary figures, have also been reviewed and adjusted according to the recommendations of the journal.

2. We note that you have indicated that data from this study are available upon request. PLOS only allows data to be available upon request if there are legal or ethical restrictions on sharing data publicly. For more information on unacceptable data access restrictions, please see http://journals.plos.org/plosone/s/data-availability - loc-unacceptable-data-access-restrictions.

Response: Thank you for highlighting this important point, we have corrected the data availability statement to accurately describe that the data is available for review during the peer review process and will be made available without restrictions once the manuscript finishes its publication cycle. 

The scripts necessary to reproduce the results, as well as the complete metadata of the samples have been deposited in the Dryad Digital Repository (https://doi.org/10.5061/dryad.np5hqbzx2) and reviewers can access it using the link below. 

https://datadryad.org/stash/share/FZ1DbrVy1KzveU5UQ_UbjqG-IrfFSouFSpspXhdnghg

Also, the raw sequencing reads and the curated reads are available in the NCBI Gene Expression Omnibus (GEO) and Sequence Read Archive (SRA) platforms with the accession code GSE205161. Reviewers can access the data using the authorized reviewer token kxsteiucbhebbkp.

We apologize for the misunderstanding about data availability and reiterate our commitment to making the results fully reproducible and the datasets freely available using these repositories. 

b) If there are no restrictions, please upload the minimal anonymized data set necessary to replicate your study findings as either Supporting Information files or to a stable, public repository and provide us with the relevant URLs, DOIs, or accession numbers. For a list of acceptable repositories, please see http://journals.plos.org/plosone/s/data-availability - loc-recommended-repositories.

Response: There are no restrictions to the use of the data and the metadata contains no sensitive information or direct identifiers of participants. The datasets also had linkable data including the date of birth, postal code, and city of residence removed before upload. The sequencing data is available in the NCBI GEO and SRA platforms with the accession code GSE205161 and the reviewers can access it using the token kxsteiucbhebbkp. The scripts to reproduce the results starting from the raw reads as well as the contextual metadata (clinical and demographic variables) are available in the Dryad Digital Repository(https://datadryad.org/stash/share/FZ1DbrVy1KzveU5UQ_UbjqG-IrfFSouFSpspXhdnghg).

 Reviewer #1 comments to the author:

This is a retrospective study evaluating whole blood gene expression using bulk RNA-seq in a cohort of cystic fibrosis patients with samples collected closest in timing to the first isolation of nontuberculous mycobacteria. The study population included patients who did (N=12) and did not (n-30) develop NTM-PD following the first nontuberculous mycobacteria growth.

1. It is not stated in the inclusion/exclusion criteria that people on CFTR modulator therapy were excluded from the analysis. However, this is mentioned in the discussion. Please add this to the exclusion criteria. Interestingly, the majority of patients were eligible for E/T/I therapy based on having one or more copies of F508del. I presume the timing of data abstraction preceded approval for E/T/I in Canada. Clarification of these details would be helpful, given most people meeting eligibility in the US (and some other locations) are likely on highly effective modulator therapy, skewing the presented results to a population that is not representative of most of North America. Please include some discussion of this issue in the manuscript.

Response: Thank you for highlighting the relevance of using CFTR modulator therapy and its possible impact on our cohort. The use of CFTR modulator therapy was not an exclusion criterion for our study but given the limited public reimbursement for CFTR modulators in Canada, only four participants (including one who developed NTM-PD) were receiving CFTR modulator therapy (1 on ivacaftor, 2 on lumacaftor/ivacaftor, and 1 on tezacaftor/ivacaftor) when the blood sample was taken. ETI was not approved in Canada until June 2021 and the first patient was reimbursed in September 2022.

We have clarified the reasons behind the low use of CFTR modulators and expanded upon the implications of our results in terms of the limited generalizability to a CFTR modulator treated population (Page 19, lines 320 to 330)

2. Please better define the patients who did not develop NTM-PD. What percentage were intermittently colonized (NTM not recovered from every AFB culture, but rather intermittently) vs chronically colonized (NTM recovery with every AFB culture) vs spontaneously cleared infection (transient or chronic colonization, but clear after one year of negative cultures)?

Response: We decided to divide our population into those who developed or did not NTM-PD as it represents a similar classification to the one observed in the non-CF population. Among the 30 patients who did not develop NTM-PD, 22 can be classified as spontaneous clearance (12 of them cleared after a single positive culture and 10 after multiple positive cultures), 3 had chronic colonization with NTM isolated in a majority of their respiratory cultures and 5 had intermittent colonization. The classification of these patients has been added to the manuscript (Results section, lines 170 to 173) and is available in the clinical and demographic dataset uploaded to the Dryad Digital Repository. 

3. Please define the data abstraction timeframe

Response: Thank you for this suggestion, we have now added the timeframe of data collection from the clinical charts. For each participant, we collected data starting from two years before the time of the first isolation (the earliest date was May 2001) and until September 2022 for follow-up of NTM status (disease or not). We have now added this important information in the Methods section, lines 104 to 107 and Results section, lines 168 and 169.

4. What was the total length of time that patients were followed to determine if they met the criteria for NTM-PD? Some patients will be intermittently positive for years before conversion to NTM disease. Did some patients that were initially without NTM-PD found to have PD after data analysis was completed?

Response: Thank you for the appropriate observation. The follow-up time of patients was variable and went from the time of the first positive culture for Mycobacteria spp. until the patient developed NTM-PD, died, or had a lung transplant (lines 107 to 109 of the Methods section). The shortest follow-up periods were seen in patients who developed NTM-PD quickly after the first isolation of an NTM. Patients who did not develop NTM-PD were followed for a minimum of 20.7 months after the first isolation of NTM (Table 1 on Page 11 and lines 173 to 176 on the Results section).

To make sure that our participants who did not develop NTM-PD up until December 2020 remained disease free, we went back to the clinical charts and reviewed their clinical status between December 2020 (when the data abstraction was initially finalized) and September 2022. None of these patients progressed to NTM-PD in the additional follow-up period. We have included this information in the Methods section, lines 104 to 107.

5. Per the manuscript, blood sampling was not limited to a specified timeframe in relation to NTM infection, but closest to the first positive growth. Presumptively these blood samples were collected exclusively after the first NTM recovery, but that is not explicitly stated. Please clarify. Additionally, 6/12 (50%) patients were sampled after initiation of NTM-specific therapy. Additional details surrounding the timing would be helpful.

Response: Our study was a secondary analysis of an ongoing cohort, and we could not control the timing of sampling around the first isolation of an NTM. Thus, we decided to take the blood sample available closest (before or after) to the time when the participant first had a positive NTM culture. As we hypothesized that an inherently decreased expression of immune response genes was associated with the development of NTM-PD, we did not restrict the sampling window. We clarified this additional information in lines 192 to 194 on page 10 of the Results section. 

As mentioned by the reviewer, half of the patients with NTM-PD had a sample taken after starting specific antimycobacterial therapy. These patients were sampled between 2 and 79 months after starting treatment due to limitations in blood sample availability in our repository. We suspected that patients who developed NTM-PD would have a similar gene expression signature to the one seen in the non-CF population, representing intrinsic defects in immunity. Thus, we did not control for the variability in sampling time. We conducted a sensitivity analysis excluding the samples taken after the diagnosis of NTM-PD and found a similar pattern of gene set enrichment (S2 Table; Discussion section, lines 317 to 320).

These limitations in sampling and their possible impact on our results are acknowledged in the Discussion section (lines 302 to 314). We also disclose the need for prospective studies with a more systematic sampling scheme to accurately establish the temporal relationship between infection and the observed patterns of gene expression. 

6. In your analysis were there any differences between M. abscessus or M. avium patients? The n was very small, so statistical differences may not be observed. However, were trends observed? Please better define these data.

Response: We thank you for the comment and agree that CF patients affected with either Mycobacterium abscessus (MABs) or Mycobacterium avium (MAC) complexes have different characteristics. In our study, we did not separate by type of mycobacteria due to the limited sample size. However, we do observe that only those infected with either MAC or MABs in our cohort progressed to NTM-PD. Furthermore, we explored clinical and demographic differences among the participants infected with MAC (n=21) or MABs (n=12). No statistically significant differences were seen in age at baseline, baseline ppFEV1, body mass index, biological sex, genotype, or co-morbidities (diabetes or pancreatic insufficiency). We have outlined this information in the Results section (lines 190 to 192 and S1 Table).

7. Table 1. Please define “follow up time”? Is this the total time a patient was followed post first + NTM culture? Reference is missing in lines 270-273.

Response: We have now clarified what “follow-up time” means in the footnote of Table 1 (page 11). Also, we have added the missing reference in line 288 (previously lines 270-273) of the Discussion.

8. Have you compared bulk RNA-seq prior to first positive NTM isolate compared to the two groups?

Response: Thank you for your comment. As part of our quality control, we have now performed a Principal Component Analysis (PCA) separating the normalized counts according to the timing of the blood sample procurement (used for RNAseq) into three groups: before culture, after culture without NTM-PD progression, and after culture with NTM-PD progression. The representation of this PCA can be found in S3 Fig. This plot shows no graphical separation of clusters in the first and second dimensions of the PCA. To further validate this finding, we evaluated if the distance matrix of the PCA had divergence according to the window between the first NTM isolation and blood sample collection. No statistically significant differences were found among the three groups.

We have added this information to lines 213 to 217 of the Results section and the PCA plot can be seen in a newly cited S3 Fig.

---

## [Editor Report · Decision Letter 1]

15 Nov 2022

Whole blood RNA-Seq demonstrates an increased host immune response in individuals with cystic fibrosis who develop nontuberculous mycobacterial pulmonary disease

PONE-D-22-24141R1

Dear Dr. Miguel Darío Prieto,

We’re pleased to inform you that your manuscript has been judged scientifically suitable for publication and will be formally accepted for publication once it meets all outstanding technical requirements.

Within one week, you’ll receive an e-mail detailing the required amendments. When these have been addressed, you’ll receive a formal acceptance letter, and your manuscript will be scheduled for publication.

Kind regards,

Abdelwahab Omri, Pharm B, Ph.D, Laurentian University

Academic Editor

PLOS ONE

---

## [Editor Report · Acceptance letter]

1 Dec 2022

PONE-D-22-24141R1 

Whole blood RNA-Seq demonstrates an increased host immune response in individuals with cystic fibrosis who develop nontuberculous mycobacterial pulmonary disease 

Dear Dr. Prieto:

I'm pleased to inform you that your manuscript has been deemed suitable for publication in PLOS ONE. Congratulations! Your manuscript is now with our production department. 

Kind regards, 

on behalf of

Dr. Abdelwahab Omri 

Academic Editor

PLOS ONE